# Multi-proxy analysis of El Olivar camelids (1,090-1,440 cal AD): Evaluating the presence of llamas (*Lama glama*, Linnaeus 1758) in the Semiarid North of Chile before the arrival of the Inca

Patricio López Mendoza[1*☉], Paola González[2☉], Michael V. Westbury[3☉], Daniela Saghessi[4☉], Lucio González Venanzi[4☉], Benito A. González[5‡], Juan C. Marín[6‡], Bárbara Rivera[7‡], Marta Valenzuela[8‡]

**1** Museo de Historia Natural y Cultural del Desierto de Atacama, Santiago, Chile, **2** Fundación El Olivar, Santiago, Chile, **3** Globe Institute, University of Copenhagen, Copenhagen, Denmark, **4** División Arqueología, Facultad de Ciencias Naturales y Museo, Universidad Nacional de La Plata, La Plata, and, Consejo Nacional de Investigaciones Científicas y Técnicas (CONICET), Argentina, **5** Laboratorio de Ecología de Vida Silvestre, Facultad de Ciencias Forestales y de la Conservación de la Naturaleza, Universidad de Chile, Santiago, Chile, **6** Departamento de Ciencias Básicas, Universidad del Bío-Bío, Concepción, Chile, **7** Independent Researcher, Santiago, Chile, **8** Páres y Álvarez Gestión Ambiental S.A., Santiago, Chile

☉ These authors contributed equally to this work.
‡ BAG, JCM, BR and MV also contributed equally to this work.
\* patriciolopezmend@gmail.com

## Abstract

To evaluate the presence of domesticated camelids in the Semiarid North of Chile (29°S) before the arrival of the Inca, we utilized a multidisciplinary approach to analyze 57 South American camelids that were part of the funerary contexts of the El Olivar site, dated between 1,155 and 1,538 cal AD and associated with the Diaguita Culture. The analyses included osteometric data, age profiles, sex estimation, genetic analysis, identification of pathologies, isotopic analysis and dental calculus analysis. The results indicate a higher frequency of juvenile-adult and adult animals, together with a relatively similar proportion of males and females. Osteometric analysis allowed us to identify the individuals as belonging to the "large group" which consists of both llama (*Lama glama*) and guanaco (*Lama guanicoe*), while genetic analysis indicates that the camelids from El Olivar are most closely related to *Lama glama* and the wild subspecies *Lama guanicoe cacsilensis*. Isotopic analyses suggest the consumption of a mixed diet of $C_3$ and $C_4$ plants, following the pattern seen in domestic camelids from the central-southern Andes. Dental calculus analyses indicate anthropic management through the provision of previously cooked food to two camelids with polydactyly. Taken together, we provide the first solid evidence of domesticated camelids (*Lama glama*) in the Semiarid region of Chile, prior to the influence of the Inca.

**Data availability statement:** All relevant data are within the manuscript and its Supporting Information files. The archaeological samples are currently deposited at the Museo Arqueológico de La Serena (La Serena, Chile) and, during the study, were located in the laboratories of the El Olivar Archaeological Project (Santiago, Chile).

**Funding:** Work funded by the El Olivar Archaeological Project. The funders had no role in study design, data collection and analysis, decision to publish, or preparation of the manuscript.

**Competing interests:** The authors have declared that no competing interests exist.

## Introduction

Information regarding domesticated camelids in the Semiarid North of Chile, an ecological and cultural zone that extends from 27° to 32° south latitude, is limited [1,2]. Studies on the domestication process in this semi-desert zone have focused on a broad temporal scales, uncovering changes in the sizes of camelids from the Late Archaic (3,000–0 BC) to the Late Period or Inca influence in the area (1,400–1,470 AD) [1–5]. Changes in size of animals under human control can arise from responses to changes in their feeding behavior, reduced mobility, availability of water, frequency of certain pathologies, and lower pressure from predators, among other factors [6]. Archaeological studies referring to the identification of domesticated camelids are largely based on osteometric analysis and have demonstrated a high variability in sizes of both domesticated and wild taxa [7,8]. In the case of genetic studies, inter-specific gene flow complicates the separating of taxa [see 9–13, among others]. Furthermore, while approaches related to diet - such as isotopic ecology - have provided key data on hunting and grazing practices in the central-southern Andes [14], they are dependent on detailed phytogeographic and paleoecological studies.

Here, we utilize the study of different proxies such as osteometry, paleopathology, genetics, isotope, and plant microremains in dental calculus of camelids that were part of the funerary contexts of El Olivar, an extensive necropolis located on the northern bank of the mouth of the Elqui River (29°S). By simultaneously considers several levels of analysis and degrees of taxonomic resolution, our multiproxy analysis provided greater resolution than those based on a single line of evidence [15]. The funerary contexts of El Olivar, whose evidence is associated with the Diaguita Culture, cover a time span of approximately 380 years and are concentrated between 1,155 and 1,538 years cal AD. The placement of complete animals in the funerary contexts as protectors or psychopomps of the deceased allowed us to obtain a large number of controlled data per individual, avoiding the bias of an averaged record as is often the case with archaeological samples.

The multiproxy analyses carried out addressed the following research objectives: **(a)** identify whether the camelids used as companions in the funerary contexts of El Olivar correspond to domestic and/or wild taxa, **(b)** evaluate the level of taxonomic resolution available to each proxy used, and **(c)** discuss these results in the socioeconomic context of the Diaguita communities of the Elqui basin. Since there is a direct connection between the semi-desert belt and the central valleys of Chile and a continuous connection with the Atlantic slope of the Andes, our results have macro-regional implications. This connection, established through artifactual evidence and stylistic analysis, has recently expanded to include the movement of animal and plant species, including those that are domesticated such as dogs [16].

### On the domestication of camelids: Problems and research questions for the Semiarid north of Chile

The Semiarid North is shaped from north to south by three main river basins: Elqui (29°S), Limarí (30°S), and Choapa (31°S), and framed within a transitional climate between the absolute desert of the Arid North and the Mediterranean environments of

Central Chile. With the appearance of the first evidence of ceramics and horticultural practices, the Early Ceramic Period in the Semiarid is defined from the year 1–800 AD. In the Middle Period, the current chronological-cultural sequence places the Ánimas Cultural Complex between 800–1,200 AD; however, this periodization is based on few absolute dates [17]. Recent research has consistently obtained dates for camelid burials with Ánimas vessels that extend into the 14th and 15th centuries AD [18,19]. With the advent of the Diaguita Culture and its different phases, the so-called Late Intermediate Period is defined, which spans approximately 1,000–1,400 AD and which, under Inca influence from 1,470 AD until the arrival of the Europeans, gives way to the Late Period. In the case of the Diaguita communities, their settlements are located on river terraces, within what has been defined as a nucleated-dispersed pattern and residential units of sedentary communities [20,21]. Their economy was broad-spectrum, characterized by crops of *Zea mays* and *Chenopodium quinoa*, vegetable gathering, hunting of different animal resources and collection of mollusks on the coast [21]. Finally, within the Diaguita world, the production of polychrome ceramics stands out and, in many cases, anthropomorphic figures are represented wearing clothing such as shirts or blankets, although the climatic conditions of the Semiarid do not allow the conservation of animal and vegetable fibers [22].

Regarding the studies on camelid domestication, for Elqui, the area where El Olivar is located, Castillo et al. [23] assigned the camelids associated with the funerary contexts of Plaza de Coquimbo dated to 800–1,200 AD to domestic llamas (*Lama glama*). Later, an osteometric and discrete morphological trait re-evaluation carried out by Becker and Cartajena [3] indicated that at least 10 of the animals were wild camelids (guanacos, *Lama guanicoe*), leaving doubts about the use of domesticated animals. Finally, during the Inca influence, the presence of llamas throughout the Semiarid is unquestionable based on zooarchaeological evidence, especially of animals used for cargo transport [2,4,5].

In the case of the Limarí basin, Francisco Cornely [24] defined the El Molle Cultural Complex (1–800 AD) as semi-sedentary groups, with agricultural and livestock practices. Camelid breeding practices of early ceramic groups was also assumed by Niemeyer et al. [25] when they pointed out that the Molles were livestock groups based on the location of plains sites, together with the representation of possible llamas in ceramic vessels and rock art. However, Becker and Cartajena [3] indicated that, for the Elqui area and probably also in the southern basins, the presence of llamas is not entirely clear for that period. Finally, for the Choapa basin, studies indicate that during the Early Ceramic Period there are no records of domesticated camelids [1], and that the llama was introduced during the Inca influence [1,26]. In the Mauro Valley (31°S), based on osteometric and isotopic studies at sites spanning 8,000 years, a direct correlation is observed between changes in size and diet, proposing a foreign introduction of domesticated camelids from the Late Intermediate Period [2,5]. On the other hand, Troncoso et al. [27] pointed out that research in the Choapa basin does not support the presence of llamas in pre-Incan archaeological contexts, while for the Elqui Valley the evidence is not conclusive. However, Cristian Becker [1] pointed out the record of llamas for the so-called Diaguita III phase for the Choapa basin.

The following research questions arise from this brief review: What was the baseline date of the appearance of the first domesticated forms of camelids in the Semiarid North of Chile? Is this first appearance explained by a local process of domestication and/or the result of a foreign introduction? Did the domesticated species have zootechnical functions similar to those in more northern areas? Do these zootechnical functions translate into morphotypes distinguishable from wild species such as the guanaco? Regarding these last two questions, interpretations of the domestication process depend largely on which of the following scenarios explain the domestication process of South American camelids: **(a)** the guanaco (*Lama guanicoe*) would be the ancestor of the llama (*Lama glama*) and the alpaca (*Vicugna pacos*), **(b)** the guanaco would be the ancestor of the llama, and the vicuña (*Vicugna vicugna*) of the alpaca, **(c)** the guanaco would be the ancestor of the llama and that the latter would have crossed with the vicuña giving rise to the alpaca and, finally, **(d)** there would have been an extinct ancestor of the llama and another of the alpaca [28]. If the llama descends from the guanaco, the effect at the osteological level observed in a temporal sequence would translate into an average decrease in the measurements belonging to the large camelids, together with the increase in the standard deviation due to the greater range of values [6]. As the domestication process is consolidated, the camelids would begin to increase in size again and

exceed that of the agriotype, which translates into an increase not only in the lower, but also in the upper range in both size groups [6].

This model, initially postulated for European artiodactyls and evaluated for areas of early domestication of camelids, such as the edge of the Salar de Atacama in northern Chile [6], is dependent on a large set of samples and a wide time scale. However, it does not account for the diversity of later zootechnical functions, such as wool production, meat production, and the use of camelids for transportation. Moreover, recent genetic studies indicate extensive hybridization of both llama and alpaca since the European conquest, along with multiple domestication centers. This calls into question the reliability of osteometric analysis from contemporary samples [13,29]. Additionally, management and care practices have evolved, affecting the amount and nature of animal reproduction, the continuation of hunting to spare domesticated animals, corral placements, induced feeding, transhumant grazing, and the social/symbolic value placed on specific phenotypes. Consequently, the study of llamas and alpacas demands a multifaceted analytical approach that considers changes in size, diet, pathology identification related to breeding and reproduction, and the contextual relationships at both micro and macro-regional scales through time.

## El Olivar

El Olivar is located in the city of La Serena (29°S-71°W), near the mouth of the Elqui River on its northern bank and approximately 2 km from the coastline (Fig 1a,1b and 1c). The site is made up of a large funerary complex and residential sector within an area of approximately 40 hectares [30,31] (Figs 2a,2b and 3a). Recent excavations focused on a polygon 380 m long and 50 m wide, where eight funerary areas were defined, in addition to 29 residential areas. The $^{14}$C dates ($n=66$, see S1 File) range between approximately 1,090 and 1,455 cal AD (Fig 3b). Within this set of dates, a period between 1,190–1,250 cal years AD is distinguished, with burials that present one or two articulated camelids, without offerings of ceramic vessels, although with the occasional presence of personal gold ornaments, copper and tin alloy

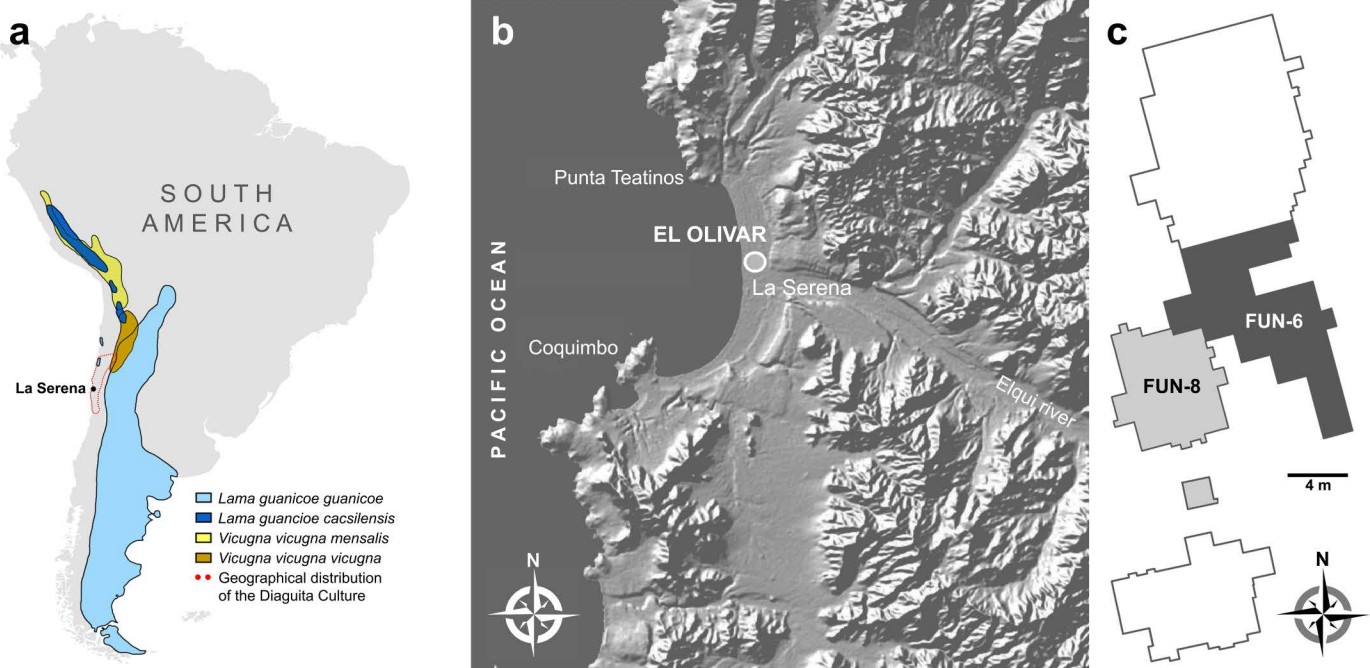

**Fig 1.** **(a) Location of the city of La Serena and distribution of wild South American camelids and archaeological evidences of the Diaguita Culture in Chile, (b) location of the El Olivar Archaeological Sector in La Serena, and (c) Plan view of sectors FUN-6 and FUN-8.** Figure 1b was created using images from Natural Earth (https://www.naturalearthdata.com).

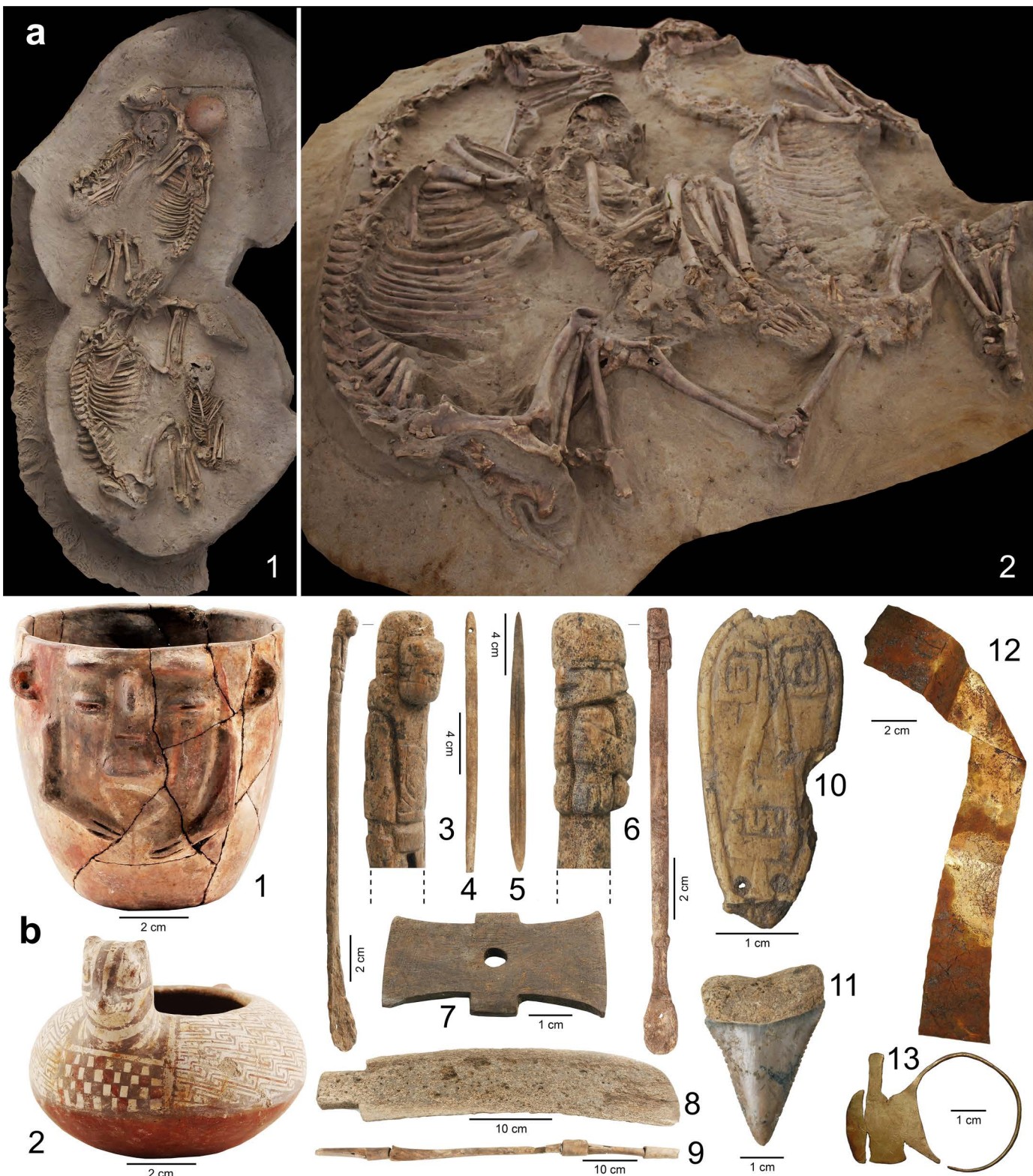

Fig 2. (a) View of two types of funerary contexts from the El Olivar site with complete camelids *in situ*: 1. Context with a camelid associated with human remains, and 2. Context with two camelids associated with a human individual, (b) Artifacts associated with the funerary

**contexts: 1. Anthropomorphic vessel, 2. Zoomorphic vessel, 3. Spatula with anthropomorphic motif, 4. Needle, 5. Harpoon stem, 6. Spatula with anthropomorphic motif, 7.** *Tortera* **for textile work, 8. Shovel made of cetacean rib bone, 9. Flute made of bird and camelid bones, 10. Anthropomorphic figure, 11. Fossil tooth of** *Carcharodon carcharias* **arranged as an offering, 12. Gold ribbon, 13. Zoomorphic gold ring.** Photographs taken by Paola González and used with permission.

artifacts, in addition to bone artifacts. Humans showed a diet with a significant contribution of marine resources, regardless of their funerary pattern (with or without camelids), while the consumption of $C_4$ plants (probably corn and/or amaranth) was also part of the diet, being especially important during childhood [19].

## Materials and methods

The analyses carried out for each camelid are detailed in Table 1. The sample set corresponded to all the camelids recovered in each funerary context, although it was not possible to carry out the all analyses on each of them due to factors such as the age of the animals, their integrity or preservation and the resources allocated for each sample.

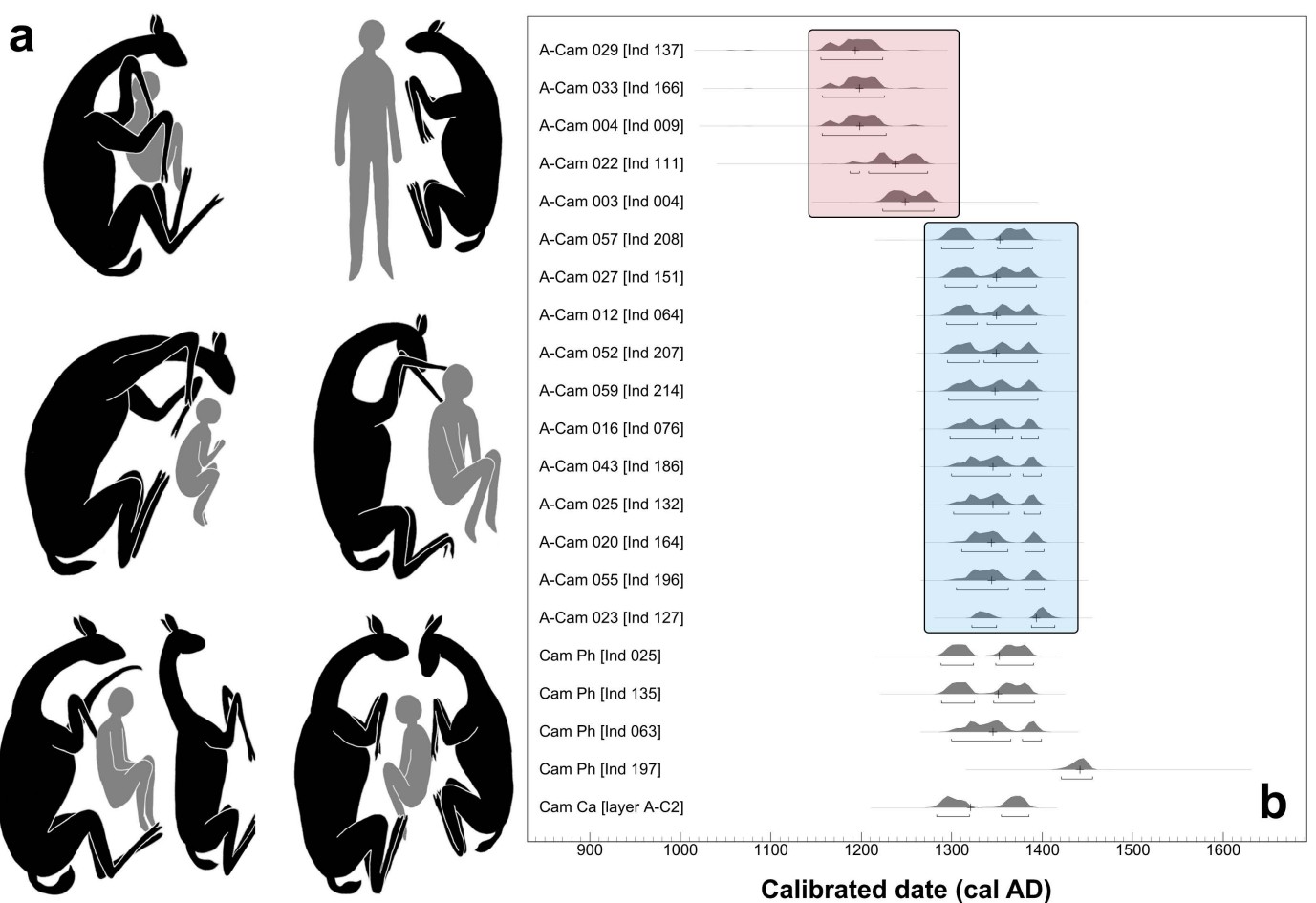

**Fig 3. (a) Diagrams of the different burial positions of camelids and humans from the funerary contexts of El Olivar; and (b) calibrated dates of articulated camelids (A-CAM) and camelid phalanges (Cam-Ph) included as burial offerings.** The red box encircles early burials with camelids, while the blue box encircles late burials of the same type. The graph includes the date of a camelid calcaneus (Cam-Ca) obtained from layer A-C$_2$. Date calibration techniques detailed in González et al. [19].

**Table 1. Camelids analysed, detailing their sex, age and analyses performed for each individual.**

| Camelid | Sex | Class (age) | Subclass (Age) | ¹⁴C | OA | GA | DC | SI |
|---|---|---|---|---|---|---|---|---|
| 1 | Female | Adults | Subclass A7 (9–10 years) | No | Yes | Yes | No | No |
| 2 | Indeterminate | Juvenile | Subclass J1 (12–19 months) | No | No | No | No | No |
| 3 | Female | Adults | Subclass A4 (6–7 years) | Yes | Yes | Yes | No | Yes |
| 4 | Indeterminate | Adults | Subclass A3 (5–6 years) | Yes | Yes | No | No | Yes |
| 5 | Indeterminate | Subadults | Subclass S1 (24–30 months) | No | No | No | No | No |
| 6 | Male | Subadults | Subclass S2 (30–36 months) | No | Yes | No | No | No |
| 7 | Indeterminate | Juvenile | Subclass J1 (12–19 months) | No | No | No | No | No |
| 8 | Indeterminate | Juvenile | Subclass J1 (12–19 months) | No | No | No | No | No |
| 9 | Indeterminate | Juvenile | Subclass J1 (12–19 months) | No | No | No | No | No |
| 10 | Indeterminate | Juvenile | Subclass J1 (12–19 months)-J2 (19–24 months) | No | No | No | No | No |
| 11 | Indeterminate | Juvenile | Subclass J1 (12–19 months) | No | No | No | No | No |
| 12 | Female | Adults | Indeterminate subclass | Yes | Yes | Yes | No | Yes |
| 13 | Indeterminate | Subadults | Subclass S1 (24–30 months) | No | No | No | No | No |
| 14 | Female | Subadults | Subclass A2 (4–5 years)-Subclass A4 (6–7 years) | No | Yes | Yes | No | No |
| 15 | Indeterminate | Adults | Subclass A1 (3–4 years) | No | Yes | No | No | No |
| 16 | Female | Adults | Subclass A1 (3–4 yers) | Yes | Yes | No | No | Yes |
| 17 | Female | Adults | Subclass A2 (4–5 years)-Subclass A4 (6–7 years) | No | Yes | Yes | No | No |
| 18 | Male | Adults | Subclass A2 (4–5 years)-Subclass A3 (5–6 years) | No | Yes | Yes | No | No |
| 19 | Indeterminate | Unborn | Subclass N1 (10–11 months of gestation) | No | No | No | No | No |
| 20 | Male | Adults | Subclass A2 (4–5 years)-Subclass A4 (6–7 years) | Yes | Yes | Yes | No | Yes |
| 21 | Indeterminate | Unborn | Subclass N1 (10–11 months of gestation) | No | No | No | No | No |
| 22 | Indeterminate | Juvenile | Subclass J1 (12–19 months) | No | No | No | No | Yes |
| 23 | Male | Juvenile | Subclass J2 (19–24 months) | Yes | No | No | No | No |
| 24 | Indeterminate | Juvenile | Subclass J1 (12–19 months) | No | No | No | No | Yes |
| 25 | Indeterminate | Young | Subclass C4 (6–9 months)-Subclass C5 (9–12 months) | Yes | No | No | No | Yes |
| 26 | Indeterminate | Subadults | – | No | No | No | No | No |
| 27 | Female | Adults | Subclass A2 (4–5 years)-Subclass A4 (6–7 years) | Yes | Yes | Yes | No | Yes |
| 28 | Female | Adults | Subclass A2 (4–5 years)-Subclass A4 (6–7 years) | No | Yes | Yes | No | Yes |
| 29 | Indeterminate | Subadults | – | Yes | No | No | No | Yes |
| 30 | Female | Juvenile | Subclass J1 (12–19 months)-Subclass J2 (19–24 months) | No | No | No | No | No |
| 31 | Indeterminate | Subadults | – | No | No | No | No | No |
| 32 | Indeterminate | Juvenile | Subclass J1 (12–19 months) | No | No | No | No | No |
| 33 | Female | Adults | Subclass A5 (7–8 years) | Yes | Yes | No | No | Yes |
| 34 | Male | Adults | Subclass A2 (4–5 years) | No | Yes | No | No | No |
| 35 | Indeterminate | Unborn | Subclass N1 (10–11 months of gestation) | No | No | No | No | No |
| 36 | Female | Adults | Subclass A2 (4–5 years)-Subclass A4 (6–7 yeats) | No | Yes | No | No | No |
| 37 | Male | Juvenile | Subclass J1 (12–19 months)-Subclass J2 (19–24 months) | No | No | No | No | No |
| 38 | Indeterminate | Unborn | Subclass N1 (10–11 months of gestation) | No | No | No | No | No |
| 39 | Indeterminate | Young | Subclass C5 (9–12 months) | No | No | No | No | Yes |
| 41 | Female | Subadults | Subclass S1 (24–30 months)-Subclass S2 (30–36 months) | No | No | No | No | Yes |
| 42 | Indeterminate | Young | – | No | No | No | No | No |
| 43 | Female | Subadults | Subclass S2 (30–36 months) | Yes | No | No | No | No |
| 44 | Male | Juvenile | Subclass J2 (19–24 months) | No | No | No | No | No |
| 45 | Indeterminate | Adults | – | No | No | No | No | No |
| 46 | Indeterminate | Young | Subclass C5 (9–12 months) | No | No | No | No | No |
| 47 | Indeterminate | Subadults | Subclass S2 (30–36 months) | No | No | No | No | No |

*(Continued)*

**Table 1.** (Continued)

| Camelid | Sex | Class (age) | Subclass (Age) | ¹⁴C | OA | GA | DC | SI |
|---------|-----|-------------|----------------|-----|-----|-----|-----|-----|
| 48 | Male | Adults | – | No | Yes | No | No | Yes |
| 49 | Indeterminate | Young | Subclass C5 (9–12 months) | No | No | No | No | No |
| 50 | Indeterminate | Adults | Subclass A1 (3–4 years) | No | No | No | No | No |
| 51 | Female | Adults | Subclass A7 (9–10 years) | No | No | Yes | Yes | No |
| 52 | Male | Juvenile | Subclass J1 (12–19 months) | Yes | No | Yes | No | Yes |
| 54 | Indeterminate | Juvenile | Subclass J1 (12–19 months) | No | No | No | No | No |
| 55 | Female | Adults | Subclass A2 (4–5 years) | No | Yes | Yes | No | Yes |
| 56 | Indeterminate | Adults | Subclass A2 (4–5 years) | Yes | Yes | No | No | No |
| 57 | Male | Adults | Subclass A1 (3–4 years) | Yes | No | Yes | Yes | Yes |
| 58 | Indeterminate | Adults | Subclass A1 (3–4 years) | No | No | No | No | No |
| 59 | Indeterminate | Adults | Subclass A2 (4–5 years) | Yes | Yes | No | No | Yes |

Camelids with polydactyly (51 and 57) are marked in black. OA. Osteometric analysis, GA. Genetic analysis, DC. Dental calculus, SI. Stable isotopes.

## Age calculation and sex estimation

Age was calculated based on eruption, dental wear, and the degree of epiphyseal fusion, and separated into age classes and subclasses as proposed by Kaufmann [32]. Sex was estimated based on the morphological characteristics of the pelvis and the size of the canines [32]. In the case of very fragmented specimens, it was not possible to determine the sex.

## Osteometric analysis

For the osteometric analysis, the first anterior or front phalanges of camelids older than 30 months were measured (Fig 4). The selection of the first anterior phalanges is due to the good resolution of these bones to identify size trends of the two size groups, the large (llama-guanaco) and the small (vicuña-alpaca) [8,33] and the greater number of comparative measurements of these bones in the specialized literature. From these measurements, the following analysis were made: **(a)** Scatter Plots, and **(b)** Cluster Analysis (UPGMA. Unweighted Pair Group Using Arithmetical Averages). The Scatter Plots were used to visualize, on a Cartesian plane, pairs of measurements that enable to identify size trendsetter. In the case of the UPGMA, the objective was to observe similarities or dissimilarities between archaeological samples and current reference animals of both size groups, using the Euclidean distance. For both analyses, measurements of the first anterior phalanges from various sectors of the southern Andean area were used, incorporating measurements from both the Pacific and Atlantic slopes, which are detailed in the S2 File, while the statistical data are summarized in the S3 File.

## Genetic analysis

**Data generation.** We obtained between ~100 mg bone powder from each sample using a Dremel drill piece. DNA was extracted from the bone powder and converted into Illumina sequencing libraries using a TECAN Fluent780 laboratory robot at the University of Copenhagen. Bone powder was first demineralized using an extraction buffer pre-digestion step for 30 minutes as described previously [35]. Following pre-digestion, DNA extractions were performed by combining 150 µl of demineralized material with 1.5 ml binding buffer (500 ml Qiagen PB, supplemented with 15 ml Sodium acetate 3M, and 1.25 ml 5M NaCl, phenol red, adjusted to pH = 5) and 10 µl of paramagnetic beads for 15 minutes [36].

Pelleted beads were washed twice in 450 µl and 100 µl 80% ethanol + 20% 10mM Tris-HCl, respectively, and eluted in 35 µl of 10 mM Tris-HCl + 0.05% Tween-20. USER treatment was performed by adding 2.5 µl USER enzyme and 7.5 µl water followed by an incubation at 37°C for 3 hours. Double-stranded Illumina libraries were prepared following the protocol described by Meyer and Kircher [37]. Indexing PCR was performed using 8-bp unique dual indexing with KAPA HiFi

HotStart Uracil+ according to manufacturer's recommendations with the number of PCR cycles required being first evaluated by qPCR. Amplified libraries were sent to Novogene (UK) for sequencing of ~5Gb on an Illumina Novoseq sequencing platform using paired-end 150 bp read chemistry. To generate a comparative reference dataset, we downloaded the raw sequencing reads from 80 South American camelids, with representatives from all four extant species (S4 File).

**Data processing.** We removed Illumina adapter sequences, low-quality reads (mean q < 25), short reads (<30 bp), and merged overlapping read pairs with Fastp. v0.23.2 [38]. We mapped to the alpaca reference genome (Genbank accession: GCF_000164845.3) using Burrows-Wheeler Aligner (BWA) v0.7.15 [39] and either utilizing the ALN algorithm, with the seed disabled (-l 999) and - otherwise default parameters - for the ancient individuals or the mem algorithm with default parameters for the modern individuals. We parsed the alignment files and removed duplicates and reads of mapping quality score <30 using SAMtools v1.6 [39]. We checked for ancient DNA damage using Mapdamage2 [40].

**Species identification.** To identify the potential species of the El Olivar individuals we used three different methods: **(a) Principal Component Analysis (PCA)**. We performed three different PCAs, one using all modern individuals, and one using only those from the *Lama* genus, and one using only *L. g. cacsilensis* and llama. As input, we computed genotype likelihoods (GL) with ANGSDv0.921 [41] using only the autosomal scaffolds >10Mb and the following parameters: -minmapQ 20 -minQ 30 -GL 2 -doGlf 2 -doMajorMinor 1 -rmtrans 1 -doMaf 2 -SNP_pval 1e-6 -minmaf 0.05 -skiptriallelic 1 -uniqueonly 1. In all cases we set a minimum individual threshold for the number of modern individuals +1. We converted our GL into a covariance matrix using PCAngsd [42]. **(b) Admixture proportions**. We used the same GL as input to PCAngsd but with the additional parameter of -admix. The most probable K was selected by PCAngsd based on Velicier's minimum average partial test. **(c) Ancestry proportions**. For this analysis we used admixfrog v0.7.2 (https://doi.org/10.1101/2020.03.13.990523) considering only the *Lama* species as the ancestry states, each represented by a single individual; *L. g. cacsilensis* - Cacsilensis4, *L. g. guanicoe* - Guanicoe1, and *L. glama* - Llama9. As input for the reference panel we created a multi-individual variant call file using BCFtools v1.15 (https://doi.org/10.1093/bioinformatics/btw044), specifying autosomal scaffolds >10Mb (-f) and minimum mapping and base qualities of 20 (-q 20 -Q 20). We ran admixfrog specifying a minimum read length of 30 bp, a bin size of 50kb, and otherwise default parameters. We only considered a bin if it contained more than three SNPs. This analysis was restricted to only the three individuals with the highest coverage (Camelids 1, 3 and 14). Two *L. g. cacsilensis* (Cacsilensis1 and 2) were likely mislabelled and therefore denoted as *L. g. guanicoe* throughout the analyses [29]. Due to the low coverage nature of some of our data we only included individuals with >0.001x average genome-wide coverage in (a) and (b).

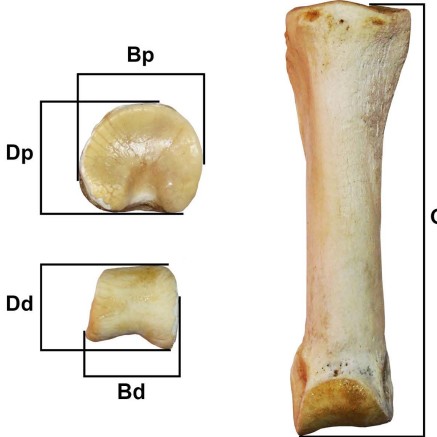

**GL:** Greatest length

**Bp:** Breadth of proximal end

**Dp:** Depth of the proximal end

**Dd:** Depth of the distal end

**Bd:** Breadth of distal end

**Fig 4. Measurements of the first anterior phalanges considered in the present analysis. Metric standards taken from von den Driesch [34].**

## Isotopic analysis

For the Semiarid North of Chile, isotopic analysis in human groups from the Late Intermediate Period show significant differences due to the enrichment of $\delta^{13}$C values compared to the Early Ceramic Period [43,44]. This change is interpreted as the result of a change in resource consumption, in which $C_4$ plants become more prevalent. For the Semiarid, this type of plants corresponds to maize (*Zea mays*), and the increase in consumption of $C_4$ plants during Diaguita times has also been observed in camelids from the Mauro Valley [2]. Therefore, through isotopic analysis in adult camelids from El Olivar, we sought to identify signals associated with the regular consumption of $C_4$ resources compatible with scenarios of human control of their diet (high $\delta^{13}$C and low $\delta^{15}$N values).

The choice of individuals ($n = 19$) was based on a selection of the entire gradient of sizes of the first anterior phalanges, previously measured for osteometric analysis detailed in Table 1. The samples used for isotopic analysis (carbon and nitrogen), were processed at the Faculty of Stable Isotopes of the University of Antofagasta and at the University of Oxford. Collagen extraction was performed using a modification of the Longin method [45] and values outside the reliable C/N range (2,9–3,6) were not included in the analysis [46]. The isotopic values are detailed in S5 File

## Dental calculus

For the dental calculus analysis, camelids 51 and 57 were selected because both presented polydactylies, so the hypothesis was raised of them possibly being domesticated forms kept near residential settlements and farming areas. To select the teeth to be sampled, the amount of dental calculus in all available teeth was first surveyed and it was decided to select the first upper molars (left and right) in both specimens because they were the teeth with the most visible calculus. A chemical-free protocol was used for sampling in order to maximize the recovery of microparticles [e.g., 47,48]. For dental calculus removal, teeth were cleaned with a brush and scraped with sterilized dental metal instruments to avoid contamination. In total, three samples were obtained (2 for Camelid 51 and 1 for Camelid 57). The obtained material was placed directly on the slide [48] and the sample was mounted with immersion oil.

This dental calculus analysis focused on the recording of vegetal microremains (phytoliths, vegetal fibers and starch grains). The description and classification of the microparticles was carried out based on the international nomenclature codes ICPN 2.0 [49] and ICSN [50]. For the taxonomic determination, metric and morphological variables were considered and reference materials and bibliography on current and archaeological cases were consulted [47,51,52]. In addition, damage to the microremains resulting from possible anthropic processing was detected, such as signs of cooking or exposure to high temperatures [53,54].

# Results

## General structure of the sample: age, sex and pathologies

The age and sex of the 57 camelids analysed are detailed in Table 1. Of the total sample, 32 camelids were not assigned to a specific sex, while 10 correspond to males and 15 to females, including animals that were pregnant at the time of their death (Fig 5a). Regarding age estimations, 23 camelids corresponded to adults, 10 to sub-adults, 15 to juveniles, five to youngs, and four were unborn. Eleven female camelids corresponded to adults, three to sub-adults, and one was a juvenile. Five male camelids were classified as adults, one as sub-adult, and four were juveniles. Seven indeterminate individuals corresponded to adults, six to sub-adults, 10 to juveniles, five to youngs, and four were unborn. This profile indicates the selection of animals of juvenile and adult ages which exceed in size the buried human individuals.

## Pathologies

A total of two camelids (51 and 57) presented polydactyly (see Fig 5b). This disorder corresponds to the presence of one or more extra toes on the feet. Polydactyly is relatively common in domesticated camelids such as

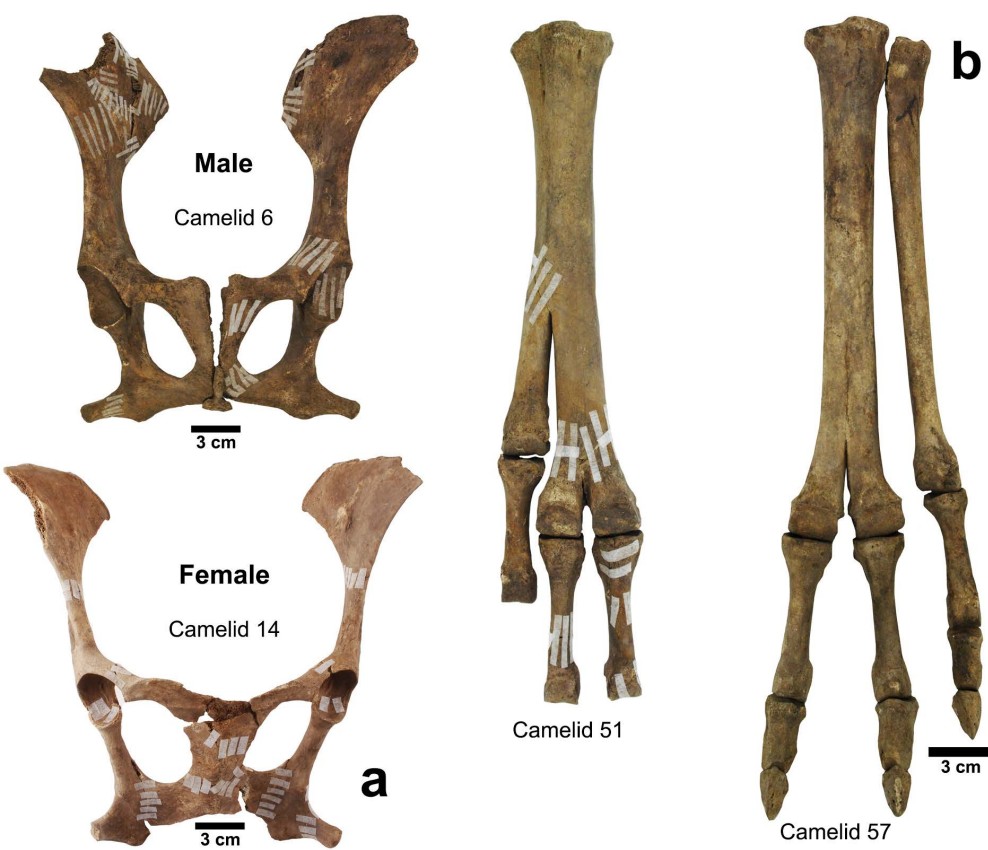

**Fig 5. (a) Pelvis of camelids 6 (male) and 14 (female); (b) Metacarpals and phalanges with polydactyly of camelids 51 and 57.**

the dromedary (*Camelus dromedarius*), llama (*Lama glama*) and alpaca (*Lama pacos*), but in exceptional cases it has been described in geographically isolated wild populations such as those in mountain areas, where genetic exchange is less or non-existent between different populations [55,56]. What is interesting in the case of El Olivar is that there are no significant biogeographic barriers that explain the presence of polydactyly. If we consider the scenario of domesticated camelids at the site, one of the causes of this pathology is the low genetic diversity due to breeding and reproduction practices that did not involve animals outside the group associated with a domestic unit or residential area [55].

## Osteometric analysis

Previous studies indicate that the northern Andean guanacos are the smallest, followed by those from intermediate latitudes such as San Juan (31°S, Argentina) and, finally, those from Patagonia [57]. Of the measurements considered as reference for the present study, the *Lama guanicoe* samples from Alto Maipo (33°S, Chile) have a size slightly larger than more northern specimens in length (GL measurement) with sizes close to the specimens from San Juan and Córdoba (31°S, Argentina). The samples from areas close to El Olivar such as Vallenar (28°S), Catamarca (28°S) and San Juan (31°S) present a high diversity of sizes in agreement with a large sample. The trend for GL, Bp and Bd measurements is relatively similar, with the highest sizes being reached in the samples from San Juan, Córdoba, Alto Maipo and Patagonia Argentina (41°S. Fig 6a).

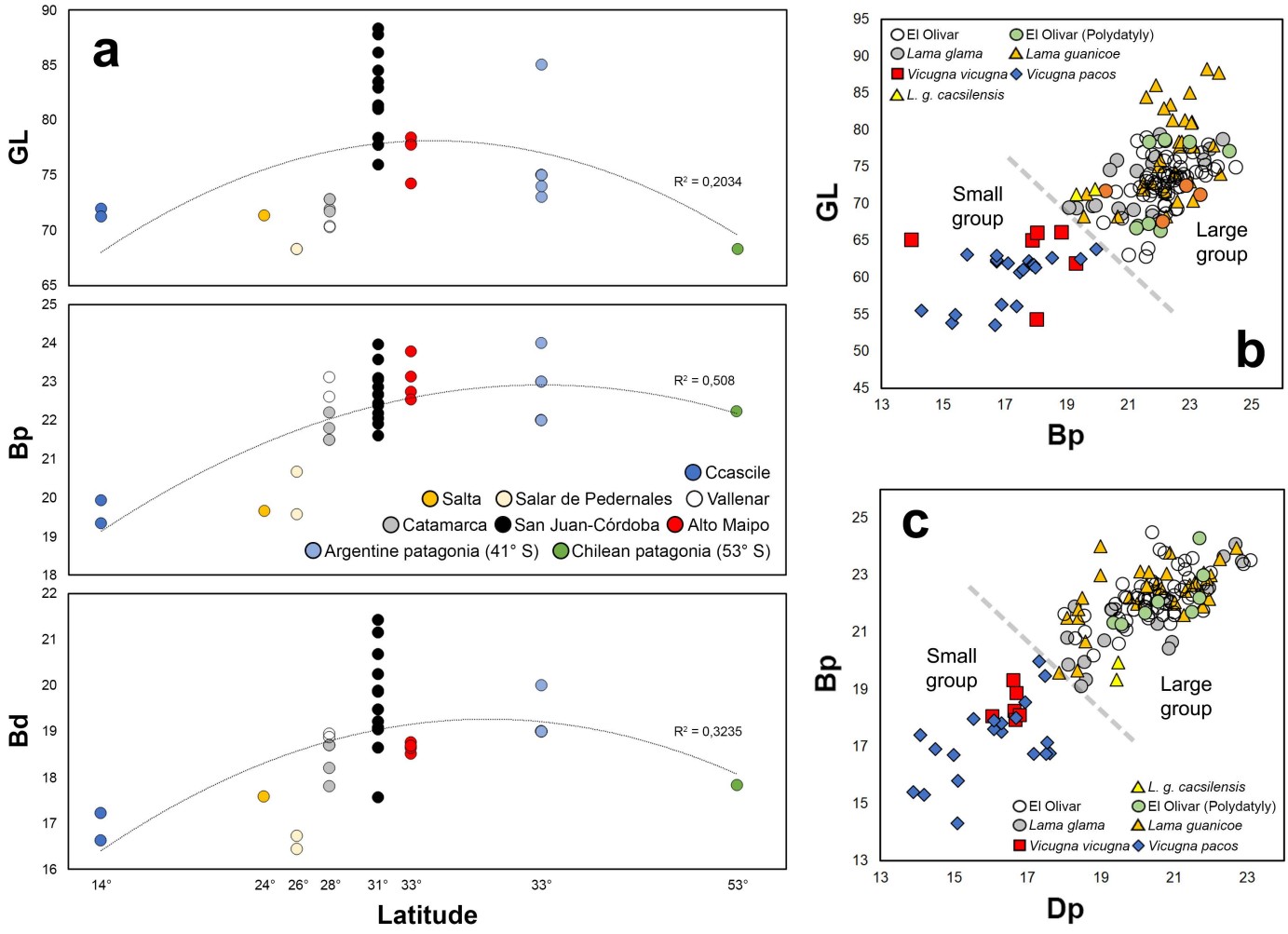

**Fig 6. (a) Distribution of GL, Bp, and Bd measurements of *Lama guanicoe* samples according to their latitudinal distribution; (b) and (c) Scatter plots representing the following measurements: GL-Bp and Bp-Dp of the first anterior phalanges.** The segmented lines in Fig 6a and 6b mark the separation between the large (*Lama guanicoe*/*Lama glama*) and small (*Vicugna vicugna*/*Vicugna pacos*) camelid groups.

From the GL and Bp measurements, a division is observed between the camelids of the large group (llamas-guanacos) compared to the camelids of the small group (alpacas-vicuñas). The specimens from El Olivar group with the first group (Fig 6b). Within the large group, the specimens from El Olivar are distributed along the entire gradient, although surpassed by reference samples of *Lama guanicoe* from Alto Maipo and Córdoba. A similar situation is observed for the Bp and Dp measurements, where both size groups are separated, although the distribution of the measurements is more heterogeneous, with the polydactyly specimens located both at the bottom and at the top of the scattering cloud of points, an aspect to be expected because part of these phalanges were not functional for locomotion (see Fig 6c). In summary, the individuals represented at the site form a heterogeneous group in their GL, Bp and Dp dimensions. The sizes of the phalanges of *Lama glama* show less heterogeneity compared to the guanaco samples and are grouped with part of the remains from El Olivar (Fig 7), with very small llamas being recorded that border on the sizes of large alpacas and vicuñas, especially the specimens from Jesús de Machaca (16°S, Bolivia).

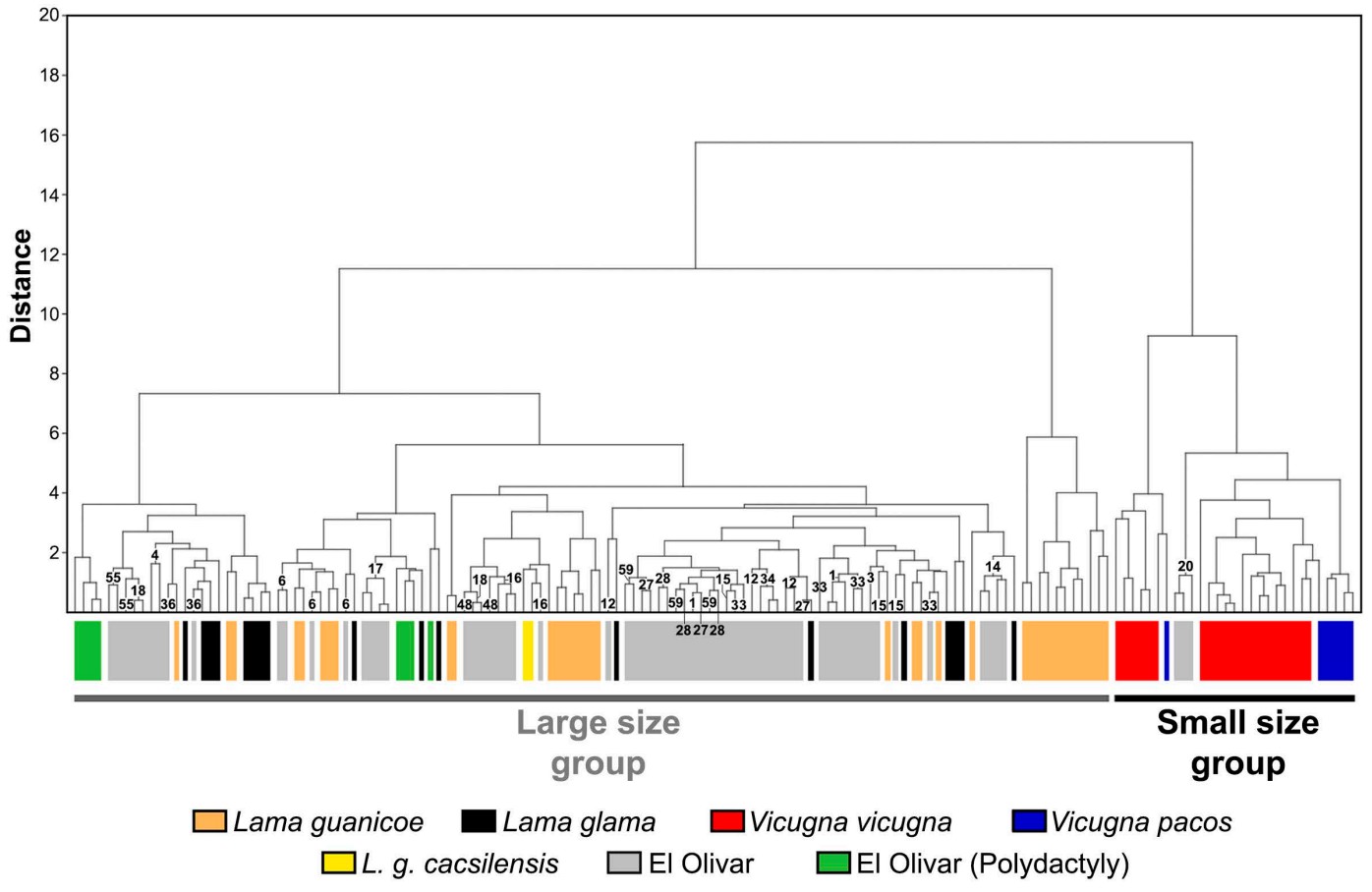

**Fig 7. Dendrogram for the first anterior phalanges of El Olivar and reference samples.** The numbers indicate each camelid at the El Olivar site.

## Genetic analyses

**Mapping.** After mapping to the alpaca reference genome, all modern samples had relatively high genome-wide coverages ranging from 8.34x to 41.11x. As for the El Olivar samples, coverages ranged between 0.0001 and 0.0118x. The ancient samples showed typical ancient DNA damage patterns with elevated levels of A-G and C-T transitions towards the read ends and short read lengths of ~50 bp.

**Species identification.** The PCA and admixture proportions all showed the El Olivar individuals to be more closely related to the *Lama* genus than to *Vicugna.* Within *Lama*, all analyses suggested a closer relationship between the El Olivar individuals and *L. g. cacsilensis/L. glama* relative to *L. g. guanicoe* (Fig 8a,b,d,e). In the *Lama* only PCA analysis (Fig 8b), it is difficult to say whether the El Olivar individuals are more closely related to the *L. glama* or *L. g. cacsilensis* as they do not fall into either cluster.

The admixture proportions show that the El Olivar individuals do have slightly more *L. g. guanicoe* ancestry than the modern *L. g. cacsilensis* or llama individuals, which could explain a slight tendency towards the *L. g. guanicoe* cluster on the PC1 axis. However, the *L. g. cacsilensis* and llama only the PCA (Fig 8c) and the ancestry proportions (Table 2) do support an overall closer relationship of the El Olivar individuals with *L. glama* than *L. g. cacsilensis*.

## Isotopic analysis

The current vegetation of the Semiarid is grouped into different zones such as the coastal zone, the intermediate valleys and the mountain zone (Fig 9a and 9b). In general, it corresponds to the so-called open shrub steppe with a predominance of $C_3$ species such as *espino* (*Vachellia caven*), annual herbs, *boldo* (*Peumus boldus*), *peumo* (*Cryptocarya alba*), *chañar* (*Geoffroea decorticans*), *molle* (*Schinus molle*), *algarrobo* (*Neltuma chilensis*), and cacti. In the mountain ranges there is an open Andean scrubland of type $C_3$ between approximately 1,000–2,000 meters above sea level with scattered shrubs such as *guayacán* (*Porlieria chilensis*) and bushes such as *Baccharis* spp., while above 2,000 meters above sea level there are $C_3$ grasses such as *Festuca* spp. and *Stipa* spp., as well as small shrubs.

Of the 19 samples analyzed, three gave C/N values outside the reliable range [46, S5 File]. The isotopic results of the camelids from El Olivar (including only those with C/N values between 2,9–3,6), indicate a homogeneous dispersion for the values of $\delta^{13}C$ ($n = 16$; mean = -16,4‰; SD = 0,7‰; range = -17,7‰ to -15,5‰) and $\delta^{15}N$ ($n = 16$; mean = 7‰; SD = 1,2‰; range 5,3‰ to 10,8‰), and a moderate negative non-significant statistical correlation (r = -0,43; p = 0,09). The $\delta^{13}C$ and $\delta^{15}N$ values, when compared with the regional isotopic ecology [44,58,59] suggest that camelids had diets with a mixed consumption of $C_3$ and $C_4$ plants (Fig 9b). Likewise, the isotopic compositions of the El Olivar specimens are similar to other coastal camelids of the Semiarid North in $\delta^{13}C$ ($n = 3$; mean = -17,3‰; SD = 2,2‰) and $\delta^{15}N$ ($n = 3$; mean = 8,9‰;

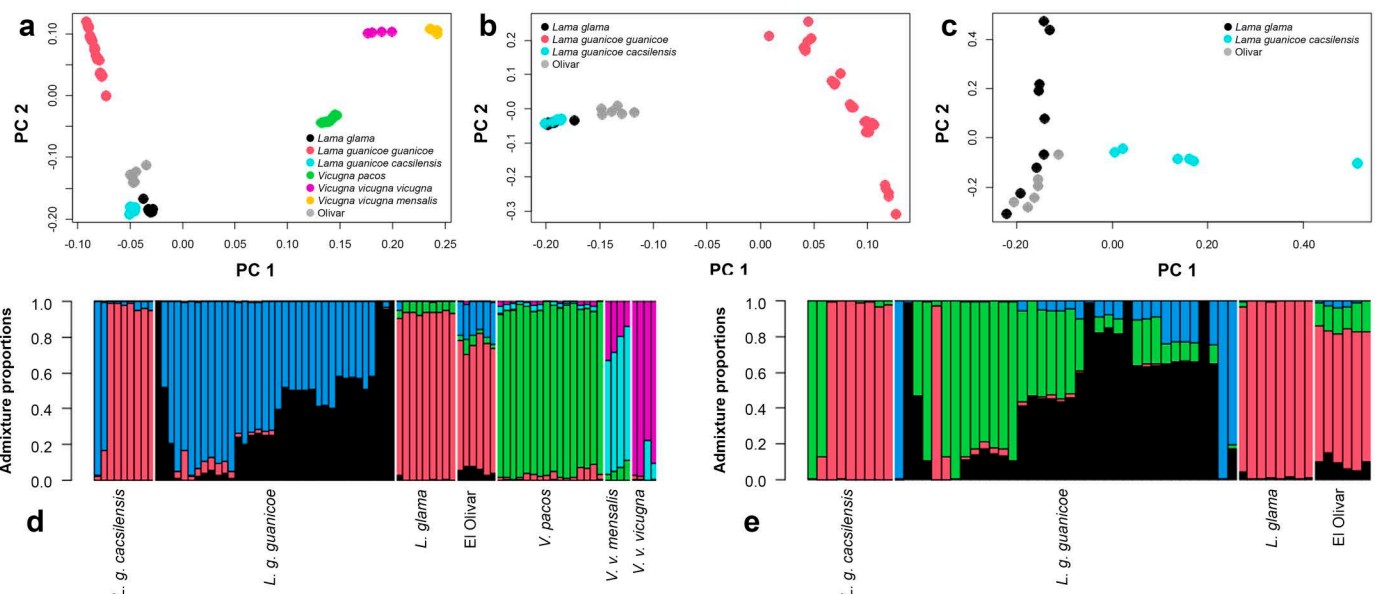

**Fig 8. PCA analysis of the El Olivar specimens with either: (a) All modern individuals, (b) just *L. glama, L. g. guanicoe,* and *L. g. cacsilensis*, and (c) just *L. g. cacsilensis* and *L. glama* individuals.** Admixture proportions taken from the GL, using a K of 6 and 4 respectively, as determined by PCAngsd and either: (d) The entire modern individual reference panel, or (e) just the Lama individuals.

**Table 2. Ancestry proportions of the three El Olivar individuals with the highest coverage (0.006-0.012x) calculated using three reference *Lama* individuals in admixfrog.**

| Sample ID | Ancestry components | | | | | |
|---|---|---|---|---|---|---|
| | *L. g. cacsilensis* | *L. g. guanicoe* | *L. glama* | *L. cacsilensis/L. glama* | *L. g. guanicoe/L. glama* | *L. g. cacsilensis/L. g. guanicoe* |
| Camelid 1 | 0,005 | 0,002 | 0,007 | 0,577 | 0,384 | 0,024 |
| Camelid 3 | 0,003 | 0,003 | 0,004 | 0,548 | 0,425 | 0,017 |
| Camelid 14 | 0,015 | 0,008 | 0,019 | 0,753 | 0,070 | 0,135 |

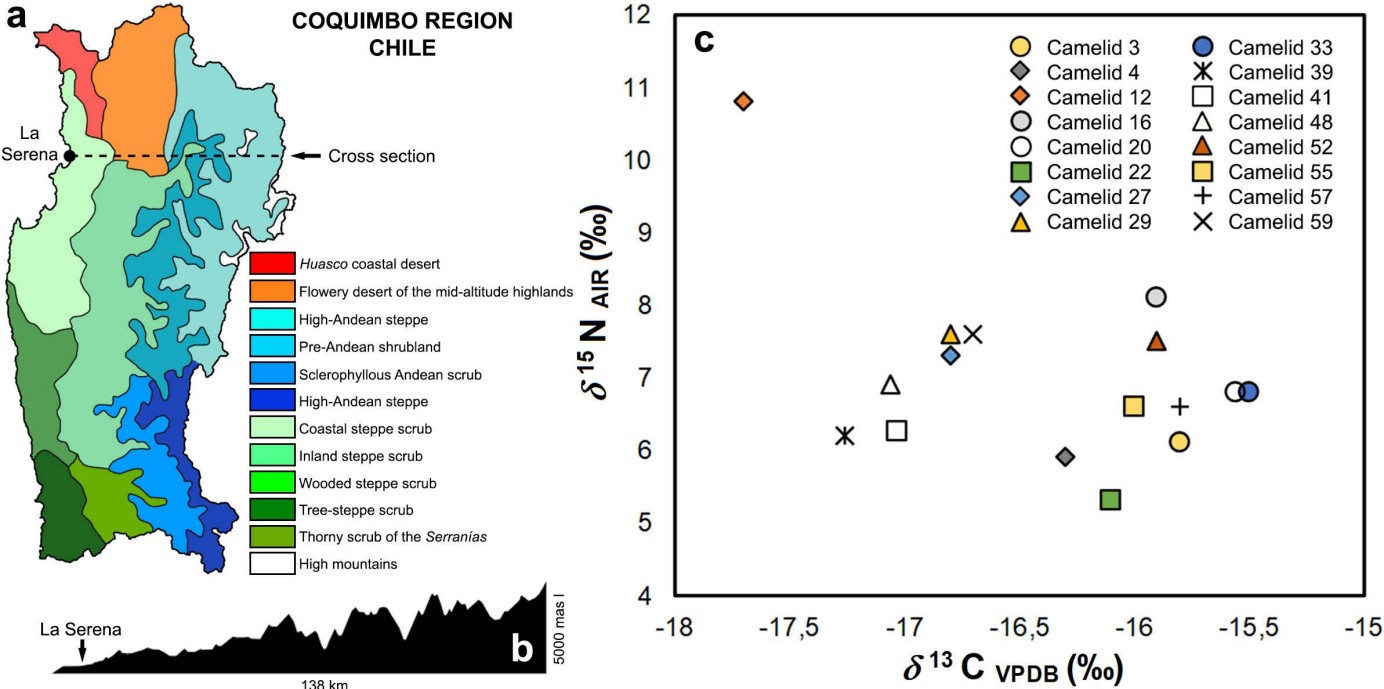

**Fig 9. (a) Plant formations of the present-day Coquimbo Region; (b) Section cut that graphs the relief of the transect from La Serena to the high mountain peaks; and (c) Bivariate graph presenting carbon and nitrogen isotope values for the El Olivar camelids.**

SD = 1,7‰) [44]. Camelid 12 from El Olivar stands out, which has a high nitrogen value, which would be related to a possible consumption of plants under the effect of aridity or of crops fertilized with terrestrial animal *guano* or dung (Fig 9c).

## Dental calculus analysis

The results of the analysis of camelids 51 and 57 (both adults and with polydactyly) indicate the presence of phytoliths (*n* = 3), vegetal fibers (*n* = 3) and starch grains (*n* = 2). Phytolith forms related to Poaceae (*n* = 1) and others not taxonomically identifiable (*n* = 2) were identified (Fig 10a and 10b). The plant fibers show tears and possibly correspond to leaves and stems of herbaceous plants (Fig 10c and 10d) [60]. In Camelid 51, a polyhedral starch grain related to *Cucurbita* spp. was recovered (Figs 10e and 10e´) and in Camelid 57, a faceted starch grain of *Zea mays* (Fig 10f and 10f´) [61,62]. Both starches showed damage to the hilum and extinction crosses, which may be due to processing of the plants prior to ingestion (cooking and grinding).

## Interpretation of the multi-proxy analysis

At the osteometric and morphological level, the camelids from El Olivar correspond entirely to the large group, made up of the llama and the guanaco. This also includes the camelids recorded in areas oriented to domestic activities at the site, as well as throughout the Semiarid of Chile [2]. Comparable reference individuals to those from El Olivar correspond to the Mauro Valley, where throughout a sequence of 8,000 years, a similar size variance of camelids is observed for the Late Intermediate Period, although during the Late Period this variance is greater. Prior to these periods, osteometric information is scarce, except for the Late Archaic (3,000–0 BC) where large camelids with a lower variance are observed, suggesting wild populations [2,5]. The lack of robust osteometric data for the period between 1 and 800 AD does not allow us to evaluate whether the domestication of camelids was a local or introduced process. The few samples from the Early

Ceramic Period indicate trends similar to those of the Archaic, with a low size variance that would rule out a local domestication process.

The assignment based on osteometric analysis to the genus *Lama* was supported by genomic analysis. All the specimens from El Olivar are most closely related to *Lama glama* and *Lama guanicoe cacsilensis*. The site's location is outside

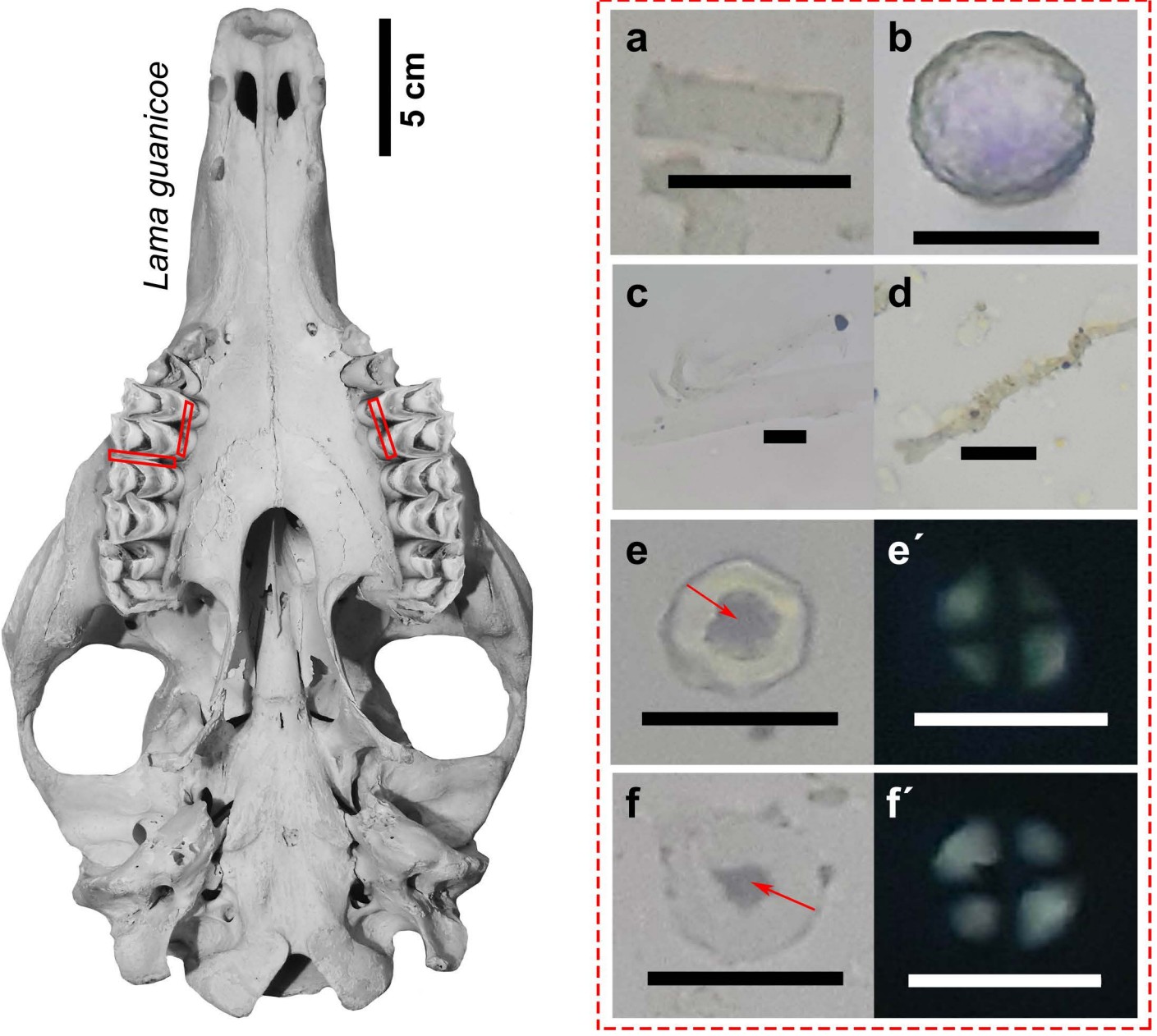

**Fig 10. Plant microremains recovered from the dental calculus of camelids 51 and 57 with markings (red rectangles) of analysed teeth (referential skull).** References: (a) Rectangular prismatic phytolith related to Poaceae; (b) Indeterminate spherical phytolith; (c) and (d) Vegetal fibres; (e-e´) Starch granule related to *Cucurbita* spp.; (f-f´) Starch granule related to *Zea mays*; and red arrows. Damage to the hilum. Scale: 20 μm.

the current distribution area of *L. g. cacsilensis*, so the hypothesis of a domesticated population based on this agriotype and transferred by humans from further north is plausible. In this regard, evidence from El Olivar such as dogs with morphologies similar to specimens from the central and south-central Andean area [16], and tin as an element present in part of the metallic artifacts from the site and absent in the geological conformation of the Chilean territory [63], indicate a direct connection of this Semiarid basin with cultural areas of northern Chile and northwestern Argentina, giving rise to a flow and exchange of information, including practices such as camelid breeding. Furthermore, the absence of Middle Period dates for El Olivar is consistent with that described for similar sites such as Plaza de Armas de Coquimbo (991–1,134 AD) and Plaza de La Serena (988–1,388 AD) [43]. Other, less direct, evidence at El Olivar and the Elqui basin indicates the appearance of a group with advanced knowledge of metallurgy (gold, copper, and copper-tin), polychrome pottery and symmetrical decoration, the emergence of a hallucinogenic complex, along with extensive textile work - judging by the iconographic information on vessels, rock art, and artifacts directly associated with textile work - which represent notable changes and differences with the southern basins, so camelid management does not necessarily extend to the entire Semiarid region of Chile.

Our initial hypothesis was that in agricultural societies with autonomous settlements such as the Diaguitas, the possible presence of domesticated forms of camelids kept close to residential settlements and maize-growing areas would be reflected in the diet of these animals. This premise is partly corroborated by the isotopic data, where mixed isotopic signatures of $C_3$ and $C_4$ plants are observed, in addition to the consumption of plants from crops possibly fertilized with *guano*. Likewise, within our multi-proxy approach, one of the strongest evidences of camelid domestication corresponds to the starches of *Cucurbita* spp. and *Zea mays* extracted from the dental calculus of camelids 51 and 57. Although these starches are scarce, which could be due to a degradation inherent to the masticatory process [44], they present sufficient features for their specific identification and evidence of possible prior processing (cooking and grinding).

Therefore, the ingestion of domesticated species by these camelids, which were essential for the groups in the region [21,64], rules out natural processes of introduction into the dental calculus, reinforces the link of these animals with human groups [see example in 65]. Moreover, this suggests the existence of little-known practices for the area, where the diet - or part of it - of some camelids was controlled by the inhabitants of El Olivar. In this context of autonomous hamlets during the Diaguita period - which reflect a high level of autonomy of extended family groups - the presence of llamas with polydactyly indicates that the breeding of these animals may not have included the mixing of llamas with livestock from neighbouring groups. It also reflects special care of animals that reached adulthood, since polydactyly causes infections in the extra toes, making walking difficult and slow, with irregular growth of the hooves of the main toes [66].

The different lines of analysis developed indicate the presence of *Lama glama* in El Olivar, leading us to the following question: What types of livestock practices were deployed during the Diaguita period in the Elqui basin? Possibly the answer lies during early historical moments, since the withdrawal of camelids towards mountainous areas was encouraged by the need for land for cultivation and the introduction of new species such as goats (*Capra aegagrus hircus*) and cattle (*Bos taurus*). The "Ordinance of 1557" prohibited grazing on low-lying cultivated land, encouraging transhumance towards the mountain range [67]. However, this transhumance in the Semiarid for historical times is in accordance with the natural productivity cycles of the environment [68], since while the pastures dry out on the coast, they grow in the mountains during December-January [67]. Thus, Aranda [69] defined the pastoral movement in some semiarid valleys as an "ascending" or "normal" type of transhumance, where the herders temporarily move their livestock up in summer, in an east-west movement. This grazing system stands out for the lack of towns in the vicinity of the summer pastures due to the inclement weather and geography. This historical transhumant management of livestock has two archaeological correlates; the first refers to the need to make east-west movements in the face of the natural productivity cycles of semiarid environments and, secondly, to the Diaguita settlement patterns. The characteristics of the semiarid environment motivate us to think of transhumant livestock movements during the Late Intermediate Period coordinated with dispersed settlements with a high degree of autonomy, an aspect that must be evaluated based on our results.

## Conclusions

The different lines of evidence used for the taxonomic analysis of the camelids from El Olivar indicate a clear presence of *Lama glama*. Answering the first two research questions initially posed in this study, the baseline date of the appearance of the first domesticated forms of camelids in the Semiarid - at least in the Elqui basin - is not clear. The scarcity of systematic data for times prior to the Diaguita culture does not allow us to completely rule out local domestication processes. However, as a future working hypothesis, it should be evaluated whether the consolidation of livestock practices occurred during the Middle Period or the beginning of the Late Intermediate Period based on the evident exchange of information with cultural areas in northern Chile and northwestern Argentina. Regarding the questions related to the zootechnical functions and morphotypes of the camelids from the funerary contexts of El Olivar, the sizes are distributed along the entire gradient of the large group formed by the guanaco and the llama. This diversity of sizes of the studied camelids may be due to the fact that, unlike other Andean areas, and prior to the entry of the Inca into the area in ca. 1,470 AD, the use of llamas was not related to the loading or deployment of large caravans.

We hypothesize, based on the abundant artefactual and iconographic evidence on vessels and rock art associated with textile work during the Diaguita [22], that the maintenance and reproduction of livestock was framed in the need for fiber production, not ruling out its use as meat producers and manure as fertilizer for crops. Livestock practices were complemented by guanaco hunting, similarly to what was observed for herders from more northern areas [14], activities, hunting and domestication, which diversified widely during Inca times.

## Supporting information

**S1 File.  ¹⁴C dating in remains of humans and camelids from the El Olivar site.** Data taken from González et al. [19].
(XLSX)

**S2 File.  Measurements of the anterior first phalanx El Olivar camelids and reference samples.** The specimen used as a standard for LSI analysis is marked in black, and the camelids with polydactyly are indicated in grey.
(XLSX)

**S3 File.  Univariate statistics of the measurements of the first anterior phalanges of camelids from the El Olivar site.** Specimens with polydactyly are included.
(XLSX)

**S4 File.  Mapping information for the modern reference panel.**
(XLSX)

**S5 File.  Carbon and nitrogen isotopic results for camelids from El Olivar.** Indicators of collagen preservation are included. Samples with poor preservation, indicated by C/N ratios, are presented in grey.
(XLSX)

## Acknowledgments

Our thanks to the entire archaeological team that participated in the field activities as a laboratory. We would like to thank Eline Lorenzen for help with the genomic wet-lab work. Patricio López Mendoza dedicates this work to the memory of Carlos López Cofré and Osvaldo Latorre Astudillo.

## Author contributions

**Conceptualization:** Patricio López Mendoza, Paola González, Michael V. Westbury, Daniela Saghessi, Lucio González Venanzi.

**Data curation:** Patricio López Mendoza, Paola González, Michael V. Westbury, Daniela Saghessi, Lucio González Venanzi, Benito A. González, Juan C. Marín, Bárbara Rivera, Marta Valenzuela.

**Formal analysis:** Patricio López Mendoza, Paola González, Michael V. Westbury, Daniela Saghessi, Lucio González Venanzi, Benito A. González, Juan C. Marín, Bárbara Rivera, Marta Valenzuela.

**Funding acquisition:** Paola González.

**Investigation:** Patricio López Mendoza, Paola González, Michael V. Westbury, Daniela Saghessi, Lucio González Venanzi, Bárbara Rivera, Marta Valenzuela.

**Methodology:** Patricio López Mendoza, Paola González, Michael V. Westbury, Daniela Saghessi, Lucio González Venanzi, Benito A. González, Juan C. Marín, Bárbara Rivera, Marta Valenzuela.

**Writing – original draft:** Patricio López Mendoza, Paola González, Michael V. Westbury, Daniela Saghessi, Lucio González Venanzi.

**Writing – review & editing:** Patricio López Mendoza, Paola González, Michael V. Westbury, Daniela Saghessi, Lucio González Venanzi.

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
