## [Decision Letter · Decision Letter 0]

28 Oct 2024

PONE-D-24-25726Multi-proxy analysis of El Olivar camelids (1090-1440 cal AD): Evaluating the presence of llamas (Lama glama, Linnaeus 1758) in the Semi-arid North of Chile before the arrival of the InkaPLOS ONE

Dear Dr. López,

Thank you for submitting your manuscript to PLOS ONE. After careful consideration, we feel that it has merit but does not fully meet PLOS ONE’s publication criteria as it currently stands. Therefore, we invite you to submit a revised version of the manuscript that addresses the points raised during the review process.

While we advise that you carefully consider all comments provided by the two reviewers, we would like to stress the particular importance of those related to clarity, replicability, and support of the paper's conclusions. Specifically, both reviewers expressed concerns regrading sampling, both as practiced and as reported. In this regard, please be sure to address R1's request for elaboration of the sampling strategies employed for each method, and R2's concerns regarding use of multiple values from single individuals and the very small sample sizes in the case of genetic sex identification and starch grain analysis. Additionally, it would be helpful to elaborate on the rationale that guided methods selection (R1). Finally, please ensure that all conclusions are well supported by the data presented.

We look forward to receiving your revised manuscript.

Kind regards,

Raven Garvey, Ph.D.

Academic Editor

PLOS ONE

Journal Requirements:

“Work funded by the El Olivar Archaeological Project.”

“Work funded by the El Olivar Archaeological Project”

4. We note that you have referenced (Cantarutti G, Cabello G. Proyecto et al. [8]) which has currently not yet been accepted for publication. Please remove this from your References and amend this to state in the body of your manuscript: (Cantarutti G, Cabello G. Proyecto et al. [Unpublished]”) as detailed online in our guide for authors

http://journals.plos.org/plosone/s/submission-guidelines#loc-reference-style .

5. We note that Figures 1 and 9 in your submission contain [map/satellite] images which may be copyrighted. All PLOS content is published under the Creative Commons Attribution License (CC BY 4.0), which means that the manuscript, images, and Supporting Information files will be freely available online, and any third party is permitted to access, download, copy, distribute, and use these materials in any way, even commercially, with proper attribution. For these reasons, we cannot publish previously copyrighted maps or satellite images created using proprietary data, such as Google software (Google Maps, Street View, and Earth). For more information, see our copyright guidelines: http://journals.plos.org/plosone/s/licenses-and-copyright.

 a. You may seek permission from the original copyright holder of Figures 1 and 9 to publish the content specifically under the CC BY 4.0 license. 

Reviewers' comments:

Reviewer's Responses to Questions

**Comments to the Author**

1. Is the manuscript technically sound, and do the data support the conclusions?

Reviewer #1: Partly

Reviewer #2: Partly

2. Has the statistical analysis been performed appropriately and rigorously?

Reviewer #1: N/A

Reviewer #2: No

3. Have the authors made all data underlying the findings in their manuscript fully available?

Reviewer #1: Yes

Reviewer #2: Yes

4. Is the manuscript presented in an intelligible fashion and written in standard English?

Reviewer #1: No

Reviewer #2: No

5. Review Comments to the Author

Reviewer #1: Summary of the research:

This is a zooarchaeological approach to the south American camelid's remains from the archaeological site of El Olivar and the Diaguita Culture located in the Semi-arid North of Chile (29°S). This site is dated between 1,090 and 1,440 cal AD. Several approaches were used here - palaeodemography determination (teeth and bone fusions), classic morphometric (phalanges)- genetic analysis (?) - Isotopic analysis of collagen bones (?) and dental tartar analysis (?). Their objective was to identify the taxonomic status from the camelids present in this area before the Inka arrival.

Overall impression:

There is a beautiful dataset and a lot of work done here but, in my mind, authors have to be more specific and to better introduce their study into the general question of the domestication of the south American camelids, with more specific/transparent analytic description for - each - approach used here. I think this is the challenge of the multiproxy analysis publications.

Moreover, after reading, I don’t have in mind what about the human population - the Diaguita Culture - and the domestication or process of domestication of the camelids here and especially because it is a funerary context? What is specific or not and compared to the rest of the Andes? it is not really clear for me after reading. Some elements from the very end of the manuscript need to be introduced before, it means starting from the introduction. In this way I think it will be easier for the reader to understand anthropological cultural and/or biological interaction related to the archaeological context here. This is the strength of a “multi proxy analysis”. I propose that authors have to better connect all the analyses between them in terms of introduction (so references too) and interpretations. Previous multi proxy analysis for other species have already shown the pertinence of the multiproxy approach to further understand human-animal relationship.

“major” and “minor” issues:

MAJOR

1. What's missing from the introduction and then from the material and method sections is why these different approaches were chosen?

a. What have they shown so far, briefly, in archaeology and camelids, that can be used to address the question of this article (so we need to go back a little further into the existing literature)?

b. and above all, how will these methods fit together to address the question? That's where the multiproxy approach comes into its own.

c. and, logically, the objectives will become clearer

For example: in this study, why are isotopes chosen? because these analyses can be used to analyze either diet, management or environmental and climatic conditions - and in relation to tartar analyses? How are they complementary, etc.?

and what does osteometry add in relation to genetics? etc. what do they want to show in relation to domestication or the domestication process?

2. This is also lacking for the justification of the sampling “strategy”,

more transparency for each method, each of which presents its own particularities and difficulties.

For example:

-phalanx osteology: mixing of anterior and posterior phalanges or not, and why?

- age calculation: be more specific? Absolute age? Age classes? When each criterion was chosen or not?

- stable isotopes: collagen evaluation criteria, etc.

Number of samples, where the measurements came from, etc. Who measured with what, etc.? This is all the more unfortunate because, if I've understood correctly, the authors have a very fine corpus corresponding to whole skeleton camelids, making it possible to lateralize, determine sex, age etc... This is very important and must be explained from the outset, especially for interpretation in the context of a multiproxy analysis, and are the analyses really cross-referenced at sample level?

Material and method description: be more specific which analyses for which purpose? With which tools? And cite references - Be more specific about all sample selection and analysis criteria for each method - The authors have done this well for genetic analyses.

3. Results section comments:

a. for osteometry, isotopes and tartar analyses, there is a mixture of background and interpretation elements that appear in the results, all of which need to be distinguished (e.g. lines 351 to 362; 416 to 419; 425 to 427; 466 to 481; 484 to 486; 501 to 502)

b. statistical analyses are also sometimes poorly described (e.g. PCA in osteometry analysis, line 48)

c. Moreover, as the choice of analyses has not been explained beforehand in the article, it is difficult to follow the interest and interaction of each analysis with the others. perhaps it would be better to target by objective... e.g. camelid size, latitudinal gradient, etc.?

d. In addition, statistical tests were carried out to confirm or refute differences observed visually only?

e. isotopic results need to be better described collagen conservation, preservation, about the quality of the values obtained and then the results range in relation to expected values etc.

etc....

4. Interpretation and conclusion sections comments:

a. For the “interpretation” section I don’t see what it does so much more than the results just before. It is not simply a matter of listing the interpretations of the results, they must be discussed with other data from the literature for camelids (or other animals if not available) and which concerns the same question.

b. The main archaeological/anthropological ideas/questions which are in the conclusion (sedentary, seasonality, connection between the area, pastoral transhumant etc.) must be integrated into the subject of the study from the very beginning, but also into the discussion to support the question, the results and the authors' interpretations. They should provide more comparisons on these same issues/results about camelids elsewhere to support or emphasize the specificity of their region.

c. Finally, attention should be paid to the interpretations in lines 516 - ‘Interpretations of the domestication process from osteometric data depend largely on the underlying assumptions surrounding the origin of domestic species’ - analyzing the domestication process with a single method in classical osteometry is impossible. The domestication process is a much more complex phenomenon and even more so in camelids. Authors must need to include more references on this topic in order to be able to use it in their discussion. How does their result cover the domestication process? On what aspects? What is the value of the multiproxy approach here in relation to the process? This is the same thing in the introduction the authors cite the domestication process but how do they analyze it in this study? It's important to be specific: studying the domestication process is not the same as looking for the presence of domestic species in an assemblage.

MINOR

- The figures are well done, but the captions need to be more precise in terms of analysis, content and supporting references, so that the reader gets all the information simply by reading them.

- English will need to be revised for some paragraph

Reviewer #2: The authors have a very broad range of proxy data for their basic question about the potential of a pastoral economy in north-central Chile. The material from El Olivar is important and relevant to their question. I have multiple concerns about the relationship between the data presented and the conclusions. In several cases, it may not be possible to answer these questions with existing data. In other cases, I feel that the authors don’t do enough to acknowledge the confounding factors that might weaken or complicate their reconstruction. In other cases, I felt that relevant archaeological data were omitted that might have been useful.

The translation of this study into English is weak since several key points are obscured by the possible differences between Spanish terms and their English cognates:

- Line 83. “partial” information. I think that “limited” would be a better term.

- Line 135 (and several other places, see line 57) “low frequency” of camelids? Compared to what values; using what measures?

- Line 156 “wild camelids vicuna” needs a fix

- Line 177 “moments” Period or occupation would be better here.

- Line 356 “non-hereditary congenital defect” ? so, not a de novo mutation? Yet the authors note: “The interesting thing about polydactyly is that one of its causes is low genetic diversity among groups of camelids” The significance of non-hereditary is unclear here.

- Line 416 “body affinities” this term suggests a degree of similarity on some level, but it isn’t clear in what forces are at work. “Affinities” suggests a degree of physical connection that the authors cannot document.

- Line 479 “corresponds to maize” as explanation for enriched carbon isotope ratios. A correspondence is a similarity but the authors seem to suggest a causal relationship. Maize consumption is one likely explanation for this observation, but other sources of enriched carbon need to be evaluated, and it would be helpful if there were a brief summary for the direct evidence for maize.

- Line 563 “Early Ceramic smokehouse” ?

Several sampling issues are apparent. This is a large sample of whole individuals in spectacular contexts, first. The osteometric data is valuable and fits well with data from multiple regions that show that multiple sizes of camelid were in one region/being used, at one time in later prehistoric periods. Yet, I feel that the use of multiple values from single individuals is inappropriate. This practice of reporting multiple values from one individual popped out when the two animals with polydactyly appeared more than 2 times in the figures (Fig 6). Such double or quadruple reporting will affect the population measures reported in comparison to other samples from disarticulated, individual bones from midden contexts.

The data on sex identification by genetic analysis is methodologically an advance, but represents such a small sample of the whole that it would have been helpful if they compared it against a random distribution. The significance of one sex or another in this setting was unclear. Similarly, the authors present age profile data without a clear comparison or suggestion of how it related to either a living herd or a sample of sacrificed animals.

The recovery and interpretation of starch grains as evidence for pastoral practices involves some tangled reasoning. The polydactylous individuals are assumed to be domesticated (though the authors cite a case of a wild animal with the same condition). The recovery of starch grains off their teeth and interpretation as cooked food is very interesting, but would be much stronger if they had recovered similar samples from other animals in the population, off ceramics, or off grinding stones. The small sample of two individuals, in the absence of related information about crop production, storage, and preparation, is premature. Similarly, though I accept the notion of llamas presence, if there are “profuse textiles” this is an important avenue to describe (fiber analysis? Isotopes, etc.?)

The discussion of camelid diversity and taxonomy is extensive but needs work. In one paragraph (line 155 and ff), the authors use the term “livestock”, “camelids”, domestic camelids” and ‘llamas” without offering much explanation for how these differ, or what the implications would be of using one term or the other. Several times they discuss an increasing or declining importance of camelids but it’s not clear if this is a geographic coverage value or if it is a comparison to the presence of other food animals. I welcome their discussion of diversity, of genetic affiliation, and the possibility of gradual movement of animal and animal-keeping traditions from the north. At the same time, I wish they would have considered the possibility that some wild animals were still being hunted, perhaps not in for the ritual purposes from this mortuary setting, but over the long haul of using and exploiting animals for food and products.

6. PLOS authors have the option to publish the peer review history of their article (what does this mean? ). If published, this will include your full peer review and any attached files.

**Do you want your identity to be public for this peer review?** For information about this choice, including consent withdrawal, please see our Privacy Policy .

Reviewer #1: No

Reviewer #2: No

---

## [Author Response · Author response to Decision Letter 1]

31 Mar 2025

Reply to Editor

Comment 1 from the Editor

While we advise that you carefully consider all comments provided by the two reviewers, we would like to stress the particular importance of those related to clarity, replicability, and support of the paper's conclusions. Specifically, both reviewers expressed concerns regrading sampling, both as practiced and as reported. In this regard, please be sure to address R1's request for elaboration of the sampling strategies employed for each method, and R2's concerns regarding use of multiple values from single individuals and the very small sample sizes in the case of genetic sex identification and starch grain analysis. Additionally, it would be helpful to elaborate on the rationale that guided methods selection (R1). Finally, please ensure that all conclusions are well supported by the data presented.

Reply to Comment 1 from the Editor

We have added more details to the methodological approaches used for each proxy. In addition, we must emphasize that much of the information, database and results are found in the supplementary files, since the main text is limited to tables and figures. We have added additional information to better justify the selection of each method, although many of these methods were selected as they have traditionally been used in the analysis of taxonomic identification of camelids and therefore comparative research is available. It is important to note that the central point of this work is the dialogue between the different analytical techniques used for a controlled sample of camelids. However, the ability and relevance of each single proxy as an indicator for the identification of domesticated species in El Olivar was also evaluated, since traditionally the approximations are osteometric, with little dialogue with other evidences.

Reviewer 1

Comment 1 from the Reviewer 1

There is a beautiful dataset and a lot of work done here but, in my mind, authors have to be more specific and to better introduce their study into the general question of the domestication of the south American camelids, with more specific/transparent analytic description for - each - approach used here. I think this is the challenge of the multiproxy analysis publications.

Moreover, after reading, I don’t have in mind what about the human population - the Diaguita Culture - and the domestication or process of domestication of the camelids here and especially because it is a funerary context? What is specific or not and compared to the rest of the Andes? it is not really clear for me after reading. Some elements from the very end of the manuscript need to be introduced before, it means starting from the introduction. In this way I think it will be easier for the reader to understand anthropological cultural and/or biological interaction related to the archaeological context here.

This is the strength of a “multi proxy analysis”. I propose that authors have to better connect all the analyses between them in terms of introduction (so references too) and interpretations. Previous multi proxy analysis for other species have already shown the pertinence of the multiproxy approach to further understand human-animal relationship.

Reply to Comment 1 from the Reviewer 1

Regarding the relationship between the human population, the bibliography on Diaguita Culture has emphasized its extensive work on ceramic vessels and at an economic level, its agricultural base has been studied in detail. When it comes to the domestication of camelids, the information is ambiguous, and this is the first work that delivers solid data to generate a scenario in which Diaguitas societies perform livestock practices. This is in the local panorama, it is very relevant because it directs the discussion towards a new social, economic and even ideological scenario, and encompasses both the societies of the Semi-Arid North and Central Chile. Furthermore, as we have seen in this manuscript, our interpretation points to an interaction with areas such as the Northwest of Argentina, which has relevant implications for both sides of the Andes.

Due to the limitations of words that Plos One offers, we cannot develop bioanthropological information too much, or detail evidence that is very diverse from the El Olivar site. Our work covers part of this evidence and more information is included in a series of articles available in specialized journals as well as in the publication process, some of which are cited in this manuscript.

In relation to the fact that “some elements at the end of the manuscript must be introduced before, and therefore, from the introduction”, we have made some changes in the manuscript related to this comment. Also, we modified sentences to better understand the interaction between each proxy.

Comment 2 from the Reviewer 1

1. What's missing from the introduction and then from the material and method sections is why these different approaches were chosen?

a. What have they shown so far, briefly, in archaeology and camelids, that can be used to address the question of this article (so we need to go back a little further into the existing

literature)?

Reply to Comment 2 from the Reviewer 1

We have extended the text to include more details about each of the approaches used. In the event that Reviewer 1 asks us to develop the relationship between animals and humans in greater depth, we consider that this exceeds the central objective of the manuscript too much, which is to test different proxy to evaluate the presence of domesticated animals in El Olivar and in Diaguita Culture. This is the first study developed in Latin America that integrates osteometric, genetic, isotopic and dental calculus analyses in archaeological camelids. For this reason, we have few references in the same area of study and in nearby areas.

Comment 3 from the Reviewer 1

b. and above all, how will these methods fit together to address the question? That's where the multiproxy approach comes into its own.

c. and, logically, the objectives will become clearer

For example: in this study, why are isotopes chosen? because these analyses can be used to analyze either diet, management or environmental and climatic conditions - and in relation to tartar analyses? How are they complementary, etc.?

and what does osteometry add in relation to genetics? etc. what do they want to show in relation to domestication or the domestication process?

Reply to Comment 3 from the Reviewer 1

On the subject of the selection of certain analyses, we have reformulated the paragraphs of the “Isotopic Analysis” section of the “Materials and Methods” section to clarify the selection criteria for the analysis of stable isotopes: “The current vegetation of the Semi-Arid is grouped into different areas such as the coast, the intermediate valleys and cordilleran zone (Fig. 9a and 9b). In general, it corresponds to the so-called open shrub steppe with a predominance of C3 species such as espino (Acacia caven), annual herbs, boldo (Peumus boldus), peumo (Cryptocarya alba), chañar (Geoffroea decorticans), molle (Schinus molle) and algarrobo (Neltuma chilensis), among others, and is also registered cacti. In the mountain ranges there is an open Andean forest of type C3 between approximately 1,000 and 2,000 masl with scattered shrubs such as guayacán (Porlieria chilensis) and shrubs such as Baccharis spp., while above 2,000 masl highlights C3 grasses such as Festuca spp. y Stipa spp., in addition to small shrubs.

Isotopic analyzes in human groups from the Late Intermediate Period, show significant differences due to the enrichment of δ13C values in comparison to the Early Ceramic Period [Becker et al. 2015, Alfonso-Durruty et al. 2016]. This change is interpreted as the result of a change in the diet, in which C4 plants acquire greater preponderance. For the Semiarid, this type of plants corresponds to maize (Zea mays), and the increase in consumption of C4 plants for Diaguitas moments has also been observed in camelids from Mauro Valley [López et al. 2015]. Therefore, through isotopic analysis in adult camelids from El Olivar, we sought to identify signals associated with the regular consumption of C4 resources compatible with scenarios of human control of their feeding (high δ13C and low δ15N values).

About the comment: This is also lacking for the justification of the sampling “strategy”, stable isotopes: collagen evaluation criteria, etc, the samples used to make paleodietary inferences were only those that presented the C/N relationship in the range 2,9-3,6 (DeNiro 1985), as indicated in the section “Isotopic analysis” from the “Materials and Methods” section. These values are presented in the S5 File.

Comment 4 from the Reviewer 1

2. This is also lacking for the justification of the sampling “strategy”, more transparency for each method, each of which presents its own particularities and difficulties.

For example:

-phalanx osteology: mixing of anterior and posterior phalanges or not, and why?

- age calculation: be more specific? Absolute age? Age classes? When each criterion was chosen or not?

- stable isotopes: collagen evaluation criteria, etc.

Number of samples, where the measurements came from, etc. Who measured with what, etc.? This is all the more unfortunate because, if I've understood correctly, the authors have a very fine corpus corresponding to whole skeleton camelids, making it possible to lateralize, determine sex, age etc... This is very important and must be explained from the outset, especially for interpretation in the context of a multiproxy analysis, and are the analyses really cross-referenced at sample level?

Material and method description: be more specific which analyses for which purpose? With which tools? And cite references - Be more specific about all sample selection and analysis criteria for each method - The authors have done this well for genetic analyses.

Reply to Comment 4 from the Reviewer 1

Part of the information is detailed in the Supplementary Files This includes each camelid used for the analysis carried out. We have also added a table, which details the analyzes carried out for each sample, and justifying the selection of these samples. There were limits to economic resources depending on the analysis, which is why it is not just an osteometric approach whose cost is lower, versus a genetic analysis whose value is greater.

In summary, we developed within the limits of possible words, the justification for each method used, adding a couple of bibliographic references.

Comment 5 from the Reviewer 1

3. Results section comments:

a. for osteometry, isotopes and tartar analyses, there is a mixture of background and interpretation elements that appear in the results, all of which need to be distinguished (e.g. lines 351 to 362; 416 to 419; 425 to 427; 466 to 481; 484 to 486; 501 to 502)

Reply to Comment 5 from the Reviewer 1

We have reformulated the lines mentioned by Reviewer 1.

Comment 6 from the Reviewer 1

b. statistical analyses are also sometimes poorly described (e.g. PCA in osteometry analysis, line 48)

Reply to Comment 6 from the Reviewer 1

We are confused about this comment, as there is no text in line 48. However, we have reformulated the osteometric analysis, eliminating the LSI analysis and leaving only the dispersion and dendrogram graphs.

Comment 7 from the Reviewer 1

c. Moreover, as the choice of analyses has not been explained beforehand in the article, it is difficult to follow the interest and interaction of each analysis with the others. perhaps it would be better to target by objective... e.g. camelid size, latitudinal gradient, etc.?

Reply to Comment 7 from the Reviewer 1

We have modified a large part of the text to make the use of each analytical tool more understandable. In the case of measurements, as we have noted, a large part of the studies on taxonomic identification of camelids are based on osteometric analysis. However, these studies also present limitations, so in this case, osteometric analyzes were used to observe whether El Olivar's samples correspond to camelids from the large group (llama/guanaco) or the small group (vicuña/alpaca), appearance which is detailed in the manuscript.

Comment 8 from the Reviewer 1

d. In addition, statistical tests were carried out to confirm or refute differences observed visually only?

Reply to Comment 8 from the Reviewer 1

As we noted in the previous point, we have eliminated statistical analyzes such as the LSI and PCA.

Comment 9 from the Reviewer 1

e. isotopic results need to be better described collagen conservation, preservation, about the quality of the values obtained and then the results range in relation to expected values

Reply to Comment 9 from the Reviewer 1

We have reformulated the stable isotope results section to incorporate the reviewer's suggestions: “Of the 19 analyzed samples, 3 obtained C/N values based on the reliable range [DeNiro 1985]. The isotopic results of camelids from El Olivar (including only those with C/N values between 2,9-3,6) indicate a homogeneous dispersion for δ13C values (n= 16; mean= -16,4‰; SD= 0,7‰; range = -17,7‰ to -15,5‰) y δ15N (n= 16; mean= 7‰; SD= 1,2‰; range 5,3‰ to 10,8‰), and a moderate statistical correlation not significant (r= -0,43; p= 0,09).

The δ13C and δ15N values, comparing them with the regional isotopic ecology [Falabella et al. 2007; Gil et al. 2011; Alfonso-Durruty et al. 2017] suggest that camelids have diets with a mixed consumption of C3 plants and C4 plants. In addition, the isotopic compositions of the El Olivar samples are similar (n= 3; mean= -17,3‰; SD= 2,2‰) and δ15N (n= 3; mean= 8,9‰; SD= 1,7‰) to other coastal camelids del Semiarid-North [Alfonso-Durruty et al. 2017]. Camelid 12 from El Olivar stands out, which presents a high nitrogen value which would be related to a possible consumption of plants under the aridity effect of crops fertilized with terrestrial animal feces (Fig. 9c)”.

Comment 10 from the Reviewer 1

4. Interpretation and conclusion sections comments:

a. For the “interpretation” section I don’t see what it does so much more than the results just before. It is not simply a matter of listing the interpretations of the results, they must be discussed with other data from the literature for camelids (or other animals if not available) and which concerns the same question.

Reply to Comment 10 from the Reviewer 1

We have modified the Interpretation section.

Comment 11 from the Reviewer 1

b. The main archaeological/anthropological ideas/questions which are in the conclusion (sedentary, seasonality, connection between the area, pastoral transhumant etc.) must be integrated into the subject of the study from the very beginning, but also into the discussion to support the question, the results and the authors' interpretations. They should provide more comparisons on these same issues/results about camelids elsewhere to support or emphasize the specificity of their region.

Reply to Comment 11 from the Reviewer 1

We consider that Reviewer 1 wants to develop a text that exceeds the limits that we can approach for the Plos One format. Carrying out a comparison with other regions, is not possible and is not part of the central objectives of this manuscript. To support aspects such as transhumance, we focus on historical information for the study area and the ecological conditions of the Elqui. The possible publication of our results in Plos One emphasizes the taxonomic identification of the sample from the aforementioned multiproxy approach, and we hope to better address sociocultural issues that arise from this identification.

Comment 12 from the Reviewer 1

c. Finally, attention should be paid to the interpretations in lines 516 - ‘Interpretations of the domestication process from osteometric data depend largely on the underlying assumptions surrounding the origin of domestic species’ - analyzing the domestication process with a single method in classical osteometry is impossible. The domestication process is a much more complex phenomenon and even more so in camelids. Authors must need to include more references on this t

---

## [Editor Report · Decision Letter 1]

9 Apr 2025

Multi-proxy analysis of El Olivar camelids (1,090-1,440 cal AD): Evaluating the presence of llamas (Lamaglama, Linnaeus 1758) in the Semiarid North of Chile before the arrival of the Inca

PONE-D-24-25726R1

Dear Dr. López,

We’re pleased to inform you that your manuscript has been judged scientifically suitable for publication and will be formally accepted for publication once it meets all outstanding technical requirements.

Kind regards,

Raven Garvey, Ph.D.

Academic Editor

PLOS ONE
---

## [Editor Report · Acceptance letter]

PONE-D-24-25726R1

PLOS ONE

Dear Dr. López,

I'm pleased to inform you that your manuscript has been deemed suitable for publication in PLOS ONE. Congratulations! Your manuscript is now being handed over to our production team.

Kind regards,

on behalf of

Dr Raven Garvey

Academic Editor

PLOS ONE